# Medical Professional Liability in Obstetrics and Gynecology: A Pilot Study of Criminal Proceedings in the Public Prosecutor’s Office at the Court of Rome

**DOI:** 10.3390/healthcare11091331

**Published:** 2023-05-05

**Authors:** Eva Bergamin, Annamaria Fiorillo, Vincenzo M. Grassi, Maria Lodise, Giuseppe Vetrugno, Fabio De-Giorgio

**Affiliations:** 1Department of Healthcare Surveillance and Bioethics, Section of Legal Medicine, Università Cattolica del Sacro Cuore, Largo Francesco Vito 1, 00168 Rome, Italy; eb95@live.it (E.B.);; 2Fondazione Policlinico Universitario Agostino Gemelli IRCCS, Largo Agostino Gemelli 8, 00168 Roma, Italy; 3Risk Management Unit, Fondazione Policlinico Universitario Agostino Gemelli IRCCS, Largo Agostino Gemelli 8, 00168 Roma, Italy

**Keywords:** medical malpractice, professional liability, obstetrics and gynecology, criminal trials

## Abstract

Criminal trials and claims against physicians for malpractice-related damages have increased dramatically in recent years, and, with Obstetrics and Gynecology being one of the medical specialties that is at the highest risk, we carried out a retrospective analysis aimed at examining all Obstetrics- and Gynecology-related medical professional liability prosecutions within the General Register of Criminal Records of the Rome Public Prosecutor’s Office between the years 2000 and 2014. The number of prosecutions increased steadily in the years 2000–2005, with varying trends in the following years. A total of 727 healthcare professionals were involved in criminal charges, and most prosecuted crimes were related to Articles 590 and 589 of the Italian Penal Code, followed by violations of Article 17 of Law 194/78. In most cases, filing was requested and granted without opposition. In 95 cases, an expert witness was appointed by the Court, and in 68 cases, the technical consultants of the State Prosecutor found culpable conduct. Public hospitals, private nursing homes and outpatient clinics, or private practices were mostly involved; in 45% of the cases, the physicians were hospital employees. In this setting, Italy is prepared to introduce new measures and regulations to address the issues posed by defensive medicine and charges of professional liability for healthcare providers.

## 1. Introduction

Criminal trials and claims against physicians for malpractice-related damages have increased dramatically in recent years, with far-reaching consequences: the rise in lawsuits causes concern, fear, and discontent among healthcare professionals, while also encouraging the practice of so-called defensive medicine, with its associated economic repercussions [1,2]. This phenomenon affects all countries to varying degrees: the lowest percentage values were recorded in Great Britain, Scandinavia, and Baltic and Eastern European countries (>50%), while Germany, Italy, and Iberian and Mediterranean countries take the lead (>200–500%), with the exception of France [3,4,5]. Given that the issue of medical liability affects all societies, there is a need to develop measures aimed at appropriately handling malpractice cases, which should also include more effective patient safety initiatives.

In this retrospective analysis, we examined all Ob-Gyn-related medical professional liability prosecutions within the General Register of Criminal Records of the Rome Public Prosecutor’s Office between 1 January 2000 and 31 December 2014, in order to identify specific trends and analyze judicial sequelae.

In those years, according to data from a 2012 ANIA survey (the Italian Insurance Companies’ National Association) [6], 8 million individuals were hospitalized in Italy each year, with 4% (320,000) of those discharged reporting healthcare-related injuries or illnesses. From April 2009 to September 2011, each year, approximately 329 patients died due to alleged malpractice, 223 of which were following alleged medical errors.

According to estimates from 2011, the disciplines most at risk of medical malpractice claims are Orthopedics (17.5%), Obstetrics and Gynecology (Ob-Gyn) (13.9%), General Surgery, and Ophthalmology (7.7%). Due to a rise in lawsuits, in recent years, many Ob-Gyn physicians have abandoned the specialty, or have limited their practice to the gynecological field, in an effort to reduce their chances of becoming involved in civil or criminal proceedings [7,8,9,10,11].

## 2. Materials and Methods

The present retrospective analysis was carried out by the Forensic Medicine Institute of the Catholic University of the Sacred Heart (Rome).

In this study, we examined all medical professional liability-related criminal proceedings (against both identified and unidentified individuals) that were registered within the General Register of Criminal Records of the Rome Prosecutor’s Office between 1 January 2000, and 31 December 2014. 

All the legislative references in the text refer to the Italian law (Italian Penal Code).

The following information was evaluated for each proceeding involving unidentified individuals: Prosecutor identification data, proceeding number, date of complaint, Italian Penal Code article pertaining to the alleged crime, date of entry into the General Register of Criminal Records of the Rome Prosecutor’s Office, date of filing request, and date on which the request for filing was granted. 

The following information was evaluated for each proceeding involving identified individuals: Prosecutor identification data, proceeding number, date of complaint, Italian Penal Code article pertaining to the alleged crime, whether the proceeding could be traced back to prior lawsuits against unidentified parties, number of suspects, date of entry into the General Register of Criminal Records of the Rome Prosecutor’s Office, date and number of filing requests for each suspect, date and number whenever the request for filing was granted, date and number of any remands for trial, direct court orders issued to specific suspects, date, number, and credentials of any convictions for specific suspects, and date, number, and credentials of any acquittals for specific suspects.

In addition, the medical specialty of each suspect was traced using the search engine of the Provincial Medical and Surgical Association of Rome (available at https://www1.ordinemediciroma.it/ accessed on 1 October 2022), and the search engine of the National Medical and Surgical Association (FNOMCEO, available at https://portale.fnomceo.it/ accessed 1 October 2022).

Following an initial assessment of 1921 criminal proceedings, 265 (14%) Ob-Gyn-related medical professional liability-related proceedings were identified. Through the consultation of TIAP (Computer Processing of Criminal Procedure Acts), we were able to access only 96 of the 265 selected proceedings due to organizational constraints (Figure 1).

From the analysis of the selected files (*N* = 96), using the data obtained from the REGE computer system, the following information could be computed:The trend of medical professional liability prosecutions in the field of Ob-Gyn in Rome, from 2000 to 2014;The types of crimes that were prosecuted;The numbers, genders and categories of involved professionals;The number of professionals for whom dismissal was requested;The number of professionals for whom the motion to dismiss was opposed;The number of professionals for whom the motion to dismiss was granted;The number of professionals for whom an indictment or direct subpoena was requested (and granted);The number of proceedings that were resolved with a conviction verdict;The number of proceedings that were resolved with an acquittal verdict.The number of proceedings in which a court-appointed expert witness provided technical advice and the consultants’ credentials;The time elapsed between the assignment and the filing of the technical consultancy;The number of cases in which expert witnesses demonstrated that professional behavior was causally related to the event;The outcome of proceedings in which no culpable behavior was found;The outcome of proceedings in which culpable behavior was proven;The number of criminal proceedings in which court-appointed technical consultants were required to provide clarifications and/or additions;The outcome of clarifications and/or additions;The number of proceedings for which a technical consultant was appointed by the party, as well as the consultants’ credentials;The number of proceedings involving Obstetrics and associated topics;The number of proceedings involving Gynecology and associated topics;The nature of the convictions;The type of healthcare facility involved;The role of involved professionals (university or hospital employees, freelance workers).

As part of the technical advice analysis, particular focus was placed on the technical advisors’ judgment of causation. The following criteria for demonstrating material causation were specifically considered: the (a) *conditio sine qua non* criterion, (b) chronological criterion, (c) qualitative criterion, (d) modal criterion, (e) phenomenological continuity criterion, (f) eligibility or scientific feasibility criterion, (g) epidemiological–statistical criterion, and (h) criterion of exclusion of other causes [12]. 

After determining culpable conduct, we moved on to determine whether the prerequisite for the outcome of a “counterfactual judgment” was also satisfied, and whether conduct that would have avoided the event “beyond any reasonable doubt” was identified.

Microsoft Excel was used as the computational tool to process the statistics and graphs shown below.

## 3. Results

The graph in Figure 2 shows how the number of Ob-Gyn medical malpractice cases increased steadily from 2000 to 2005, peaking in 2001 and 2002 (*N* = 16 in 2000, *N* = 11 in 2001, *N* = 25 in 2002, *N* = 22 in 2003, *N* = 29 in 2004, *N* = 26 in 2005). There was then a significant decline in 2006, which was followed by fluctuating trends until 2009 (*N* = 14 in 2006, *N* = 10 in 2007, *N* = 16 in 2008, *N* = 12 in 2009).

In 2010–2011, we observed a new increase (*N* = 22 in 2010, *N* = 20 in 2011); the trend was then interrupted in 2012 (*N* = 9), only to resume in 2013 (*N* = 20). A minor decrease was also observed in 2014 (*N* = 13).

According to the analysis of all criminal proceedings, most crimes that were prosecuted in the field of Ob-Gyn were in violation of Italian Penal Code Articles 590 (Culpably caused personal injuries) and 589 (Manslaughter) (more than 90% of the total, *N* = 174 and *N* = 72, respectively).

Article 17 L 194/78 (regulations governing maternity social protection and voluntary pregnancy termination) and the following articles of the Italian Penal Code outline the remaining criminal charges: article 328 (refusal to carry out official duties. Omission) (*N* = 1), article 373 (false expertise or interpretation) (*N* = 1), article 476 (falsity committed by a Public Official in a Public Act) (*N* = 1), articles 582 and 583 (personal injury; aggravating circumstances) (*N* = 1 and *N* = 1, respectively), article 593 (failure to assist, i.e., hit-and-run) (*N* = 1), article 640 (fraud) (*N* = 3) and article 113 (cooperation in manslaughter) (*N* = 1).

From 2000 to 2014, there were a total of 727 physicians and healthcare personnel involved in Ob-Gyn professional liability prosecutions in Rome, 66% of whom were males (*N* = 471).

Figure 3 shows a graph depicting the number of specialists involved, subdivided by category: gynecologists were the most frequently involved (*N* = 437), followed by anesthesiologists (*N* = 42), surgeons (*N* = 33), obstetricians (*N* = 28), nurses (*N* = 25), radiologists (*N* = 17), pediatricians (*N*=13) and urologists (*N* = 6). In 75 cases, the specific category could not be determined, while 51 professionals belonged to various other categories (cardiologists, internal medicine, neurosurgery, etc.).

A review of all Ob-Gyn medical malpractice proceedings revealed that filing was requested in most cases (*N* = 432, 59%). Filing was upheld, unopposed, in 278 cases (38%). Of the 432 professionals for whom the request was submitted, 146 (34%) faced opposition to filing, while in 114 cases (78%) the motion to dismiss was instead approved. An immediate call to trial was issued for 11% of professionals (*N* = 16), further investigations were conducted in 5% of cases (*N* = 8), indictment requests were submitted in 3% of cases (*N* = 5), and no data were available for one (0.68%) healthcare professional. In total, 14 of the 16 professionals who received an immediate call to trial were acquitted, one was convicted, and one was being investigated as of 2014; all healthcare practitioners (*N* = 5) who were remanded for trial received nonsuit judgments.

Thus, in general, filing was ordered for a total of 392 (54%) of the 727 healthcare professionals involved.

Direct arraignments were issued for 169 (23%) of the 727 professionals involved, and indictments were requested in 92 cases (13%). Of the 92 professionals, 83 (90%) were remanded for trial. A non-prosecution ruling was issued in 18 of these cases (21% of the indictments and 2% of the total) (*N* = 9, the fact does not constitute a crime; *N* = 4, the fact does not exist; *N* = 3, the crime was not committed; *N* = 1, insufficiency of evidence; *N* = 1, dismissal of the complaint). Two professionals accepted plea deals.

Convictions were issued for 24 (3%) of the 727 professionals involved. Sentences commissioned ranged from paying a fine and court costs to imprisonment (minimum: 2 months; maximum: 9 months).

For 165 (23%) of the 727 healthcare professionals involved, acquittals were issued for the following reasons: the crime did not exist (*N* = 46, 28%), the complaint was dismissed (*N* = 35, 21%), the crime was barred by the statute of limitations (*N* = 26, 16%), the fact did not constitute a crime (*N* = 23, 14%), the crime was not committed (*N* = 16, 10%), (*N* = 35, 21%), a faulty complaint was submitted (*N* = 12, 7%), late filing of complaint (*N* = 4, 2%), death of the defendant (*N* = 2, 1%), non-prosecutability of the act (*N* = 1, 1%).

An expert witness was court-appointed in 99% (*N* = 95) of the 96 Ob-Gyn proceedings. A panel of experts was appointed in 79% of the latter cases, while only one consultant was nominated in 22% of cases (*N* = 21). In 66% (*N* = 63) a collegial (medical examiner + gynecologist) was appointed; in 15% (*N* = 14) of the cases, on the other hand, only one medical examiner was nominated, and in 7% [7] of the cases, only a gynecologist was selected. For 11 (12%) cases, the assignment was given to non-specialists in forensic medicine or gynecology (i.e., to a panel of consultants other than forensic practitioners and gynecologists). The average time elapsed between the assignment and the filing of the review was 302 days.

The Technical Consultants of the State Prosecutor (TCSP) identified culpable conduct in 72% (*N* = 68) of cases, while in 28% of cases (*N* = 27) they did not identify culpable conduct. In the only case in which the technical consultation was not commissioned, the Prosecutor did not identify culpable conduct and the case was dismissed.

A motion to dismiss was filed in all proceedings where no culpable conduct was found. Dismissal was granted in 52% of the latter (*N* = 11), opposition to dismissal was presented in 55% (*N* = 15), clarifications were requested from the TCSP at the end of the proceedings opposing dismissal in 40% (*N* = 6), and the judgments were sustained in all cases. After opposition, 87% (*N* = 13) of the proceedings were dismissed. There was an acquittal in one proceeding and a request for an evidentiary incident in another. It was not possible to reconstruct the outcome of the opposition to dismissal in one proceeding. In total, dismissal was ordered for 88% of the criminal proceedings for which culpable conduct had not been identified.

Counterfactual reasoning was addressed by the TCSP in 29% (*N* = 20) of the 68 cases in which culpable conduct was identified, and in three of these the causal relationship to the complained-of outcome was not demonstrated. In the remaining 71% (*N* = 48), the TCSP identified exigent conduct but did not use counterfactual reasoning.

The graph in Figure 4 depicts the distribution of prosecution outcomes for the 727 professionals involved based on whether or not the TCSP used counterfactual reasoning when determining culpable conduct. In the case of counterfactual reasoning, the distribution was less varied: new investigations were carried out for 3.9% (*N* = 29) of the professionals, filings were carried out for 2.3% (*N* = 17) of the professionals, to which must be added acquittals after opposition to filing (1.5%, *N* = 11), acquittals were formulated for 1.7% (*N* = 13) of the professionals, convictions were 0.2% (*N* = 2), and remands for trial comprised 0.2% (*N* = 2).

The motivation for acquittals was as follows: *N* = 3, the act does not exist; *N* = 1, nonsuit cases; *N* = 4, statute of limitations; *N* = 2, the act does not constitute a crime; *N* = 4, defect/delay of the complaint; *N* = 1, dismissal of the complaint.

In cases where the TCSP did not apply counterfactual reasoning, 8% (*N* = 59) of professionals were acquitted, 5% (*N* = 37) were filed after opposition, 0.6% (*N* = 5) were filed without opposition, 2.2% (*N* = 16) were convicted, 0.1% (*N* = 1) have been sued, and 1.6% (*N* = 12) have been remanded for trial. The motivation for acquittals was as follows: *N* = 10, the act does not exist; *N* = 3, nonsuit cases; *N* = 5, to statute of limitations; *N* = 7, the act does not constitute a crime; *N* = 19, complaint fault/delay; *N* = 9, complaint dismissal; *N* = 5, the act has not been committed.

Clarifications were requested from the TCSP in 20% (*N* = 20) of criminal proceedings. In the remaining 76 (80%) proceedings, the TCSP confirmed the conclusions expressed in the previous review.

In 49% (*N* = 47) of cases, the parties did not appoint their own consultants; on the other hand, in 23% (*N* = 22) of cases, the party-appointed technical consultants (CTP) were medical-legal and gynecological consultants; in 15% (*N* = 14), only a medical examiner was appointed, while for 10% (*N* = 10), only a gynecological consultant was appointed. 

For 64% (*N* = 61) of the analyzed prosecutions, obstetrics was the area of interest, while the remaining 36% (*N* = 35) were related to gynecology.

The most frequent incidents in criminal cases of gynecological interest were post-operative complications of gynecological surgeries (bleeding, septic complications, pelvic adhesions), which occurred in 54% (*N* = 19) of cases; abdominal organ injuries (ureteral, bladder, and bowel injuries) occurred in 34% (*N* = 12) of cases; and surgical material retention in the abdomen occurred in 11% (*N* = 4) of cases.

In total, 33% (*N* = 20) of obstetrics cases involved postpartum complications (postpartum hemorrhage, retention of placental material, etc.); 20% (*N* = 12) of the cases involved fetal distress in labor; 16% (*N* = 10) of the cases involved intrauterine fetal death; other complications (infection, sepsis, injury) occurred in 11% (*N* = 7) of the cases; 6% (*N* = 4) of cases involved shoulder dystocia, as well as voluntary termination of pregnancy complications (*N* = 4); in 5% (*N* = 3), there were complications of extrauterine pregnancies; and in 3% (*N* = 2) there were missed ultrasound diagnoses of fetal malformation.

Convictions were equally distributed in the Gynecological and Obstetrical fields: in the former, the topics were surgical material retention and the development of bladder/ureteral injuries during gynecological surgery; in the latter, the topics were intrauterine fetal death, complications of voluntary pregnancy termination, and extrauterine pregnancy.

In 44% (*N* = 42) of the cases, the involved facility was a public hospital, while in 34% (*N* = 33) it was a private nursing home, in 12% (*N* = 11) it was an outpatient clinic or private practice, and in 10% (*N* = 10) it was a university hospital.

In 45% (*N* = 80) of the cases, the physicians involved were hospital employees; in 33% (*N* = 58), they were employees of a private nursing home; in 11% (*N* = 20), they were engaged in freelance practice; in 10% (*N* = 17), they were academics; and in 1% (*N* = 2), the role they played could not be identified.

## 4. Discussion

As previously stated, one of the medical specialties most at risk for litigations is Ob-Gyn. Still, regardless of the growing magnitude of the phenomenon, studies on the subject aimed at analyzing judicial sequelae are few and insufficiently detailed, both in Italy and elsewhere. Despite the Scientific Community’s growing interest in medical liability issues over the last decade, quantitative and qualitative data on the true extent of the phenomenon are currently unavailable.

Given that the way medical malpractice cases are handled varies from country to country, we studied the situation in Italy to get a sense of the scope of the problem and provide insight into the current state of one of the medical specialties that is most frequently the subject of malpractice claims [13]. 

Our retrospective analysis showed an increasing—although fluctuating—trend in the number of Ob-Gyn medical malpractice cases from the years 2000 to 2014, which should also be interpreted in light of patients’ more conscious awareness of their rights and improved quality of life, not to mention the sponsorship campaigns of associations and law firms that promote litigation by promising financial compensation to alleged victims of medical malpractice.

Indeed, our study showed a total of 727 healthcare professionals involved in Ob-Gyn-related medical malpractice criminal charges, the majority of whom were men (66%), as was described by Chauan et al. [10] who conducted a study aimed at describing the frequencies and outcomes of claims made against members of the Central Association of Obstetricians and Gynecologists (CAOG), and determining whether the demographic characteristics of physicians allow for the identification of those who are more likely to face litigation. In the latter study, data were gathered by sending anonymous surveys to CAOG members via e-mail. It was observed that the average age of CAOG members facing litigation was 58 years old, with 84% being males.

In accordance with the literature, our retrospective analysis showed that Obstetrics and Gynecology were most frequently involved (60%) in professional liability proceedings. Of the latter, Obstetrics was found to be the field most at risk for litigations (64% vs. 36%), primarily due to problems during labor and delivery for both the mother and the fetus [14,15,16,17,18,19,20,21,22,23,24,25,26,27]. In the gynecological setting, on the other hand, the most frequently represented events were postoperative complications of gynecological surgeries (bleeding, septic complications, pelvic adhesions) and injuries to abdominal organs during gynecological procedures (ureteral, bladder, bowel injuries). In the obstetrical field, convictions involved intrauterine fetal deaths, complications of voluntary termination of pregnancy, and extrauterine pregnancy complications, while convictions in the gynecological field focused on bladder/ureteral injuries after gynecological surgeries and the retention of surgical material in the abdomen. 

As far as the literature is concerned, Shwayder et al. [16] identified nine macro-areas of obstetric litigation: errors or omissions in prenatal screening and diagnosis, infant neurological injury, neonatal encephalopathy, stillbirth or neonatal death, shoulder dystocia, vaginal delivery after cesarean section, vaginal operative delivery, and ultrasound diagnosis. Sanders et al. [23], on the other hand, found that medical professional liability lawsuits for obstetric misdiagnoses at ultrasound accounted for most of the litigations (75%), followed by gynecological cases (10.4%). Extrauterine pregnancy and misdiagnoses of malformations at ultrasound account for 23% and 42% of all litigations, respectively, along with inaccurate fetal number assessment (11%), placenta previa (7.2%), and adnexal tumefaction (2.5%). Despite the widespread use of transvaginal ultrasonography, the authors also observed an increase in the number of cases of fetal malformation misdiagnoses (*N* = 6 in 1983; *N* = 40 in 1996) and extrauterine pregnancies (*N* = 12 in 1983; *N* = 22 in 1996).

In terms of the characteristics of the healthcare facilities involved, the study findings are supported by data from the literature that identify small city or provincial hospitals, private nursing homes, and private practices as the most vulnerable; as a result, the majority of physicians are either hospital employees, private clinic employees (who perform freelance work internally), or independent contractors.

As regards the legal implications of the 727 professionals involved in Ob-Gyn medical malpractice proceedings, in most cases (59%), filing of the claim was requested and upheld (78%), while direct arraignment was issued for 11% (*N* = 16) of the professionals involved, 14 of whom were acquitted and only 1 convicted.

In 95 out of 96 examined proceedings, an expert witness was appointed by the Court; a panel of consultants was appointed in 66% of these cases. Only one medico-legal expert was assigned in 15% (*N* = 14) of the studied cases, and only one Gynecologist was appointed in 7% (*N* = 7) of the cases. 

Culpable conduct was found in 72% (*N* = 68) of the proceedings, though counterfactual reasoning was employed in only 29%. Different outcomes in terms of judicial sequelae were observed according to whether counterfactual reasoning was applied or not.

Worthy of attention is the estimate of the number of physicians who were actually convicted and for whom liability was recognized (and, thus, the existence of the causal relationship between their conduct and the occurred event). This modest percentage is probably the result of the different criteria used to establish the existence of a causal relationship in the criminal context, which requires it to be proven with a “strong or high degree of rational credibility” or “logical probability”, or “beyond any reasonable doubt”, as stated by the United Sections.

Given the low percentage of convictions, as opposed to the high percentage of filings and acquittals, one wonders whether the time has not really come to make significant attempts to achieve reform in the field of criminal liability in healthcare. However, as we wait for new proposals and regulations on the challenges posed by defensive medicine and professional liability charges of health practitioners, one might learn from the experiences of other countries [28]:In Scandinavian countries, a system for managing the consequences of adverse events known as the “no-fault compensation system” has been promoted, which provides financial compensation for all those who demonstrate iatrogenic injury, regardless of the demonstration of censurable profiles of physicians’ behavior or structural deficiencies [29,30];The French judicial system is centered on Law 303 of 4 March 2002, which enshrines, in a nutshell, the obligation of compensation in connection to the risk associated with performance rather than the establishment of fault. The universality of the risk’s exposure and the ad hoc nature of the risk’s actual scope lead to a principle of solidarity that extends the obligation of compensation to the entire community. This is referred to as “socialization of risk”—if the liability of a professional, hospital, or healthcare institution cannot be demonstrated, the patient who is the victim of a medical accident may file a claim for damages as a form of national solidarity. However, the damage must be directly related to an act of prevention, diagnosis or treatment, and must exceed a minimum severity level determined by a specific decree;Furthermore, alternative out-of-court forms such as mediation, conciliation, and arbitration have been promoted in the international arena (United Nations, as well as some EU states: England, Austria, France, and Germany). In practice, it seeks to handle the conflict between doctor and patient using simple procedures that allow for a resolution in a short period of time and at a low cost, while avoiding the emotional and psychological stress associated with the process [31,32,33,34,35,36,37].

## 5. Conclusions

The present retrospective analysis has once again shown how the issue of medical malpractice is ever-present and is becoming a major concern, especially with regard to the field of Ob-Gyn, with an increase in the number of medical malpractice claims and physicians subject to litigations.

In this setting, Italy is prepared to introduce new measures and regulations to address the issues posed by defensive medicine and charges of professional liability for healthcare providers.

This would significantly reduce the number of medical professional liability prosecutions, resulting in work relief for Prosecutors and Judges; physicians would be able to devote themselves more peacefully to their profession, and defensive medicine behaviors would be reduced, resulting in significant savings for the National Health System.

## Figures and Tables

**Figure 1 healthcare-11-01331-f001:**
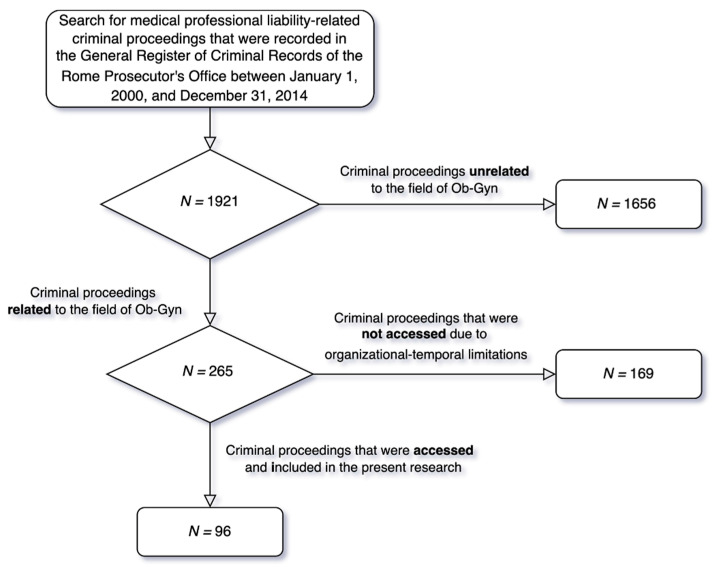
Flowchart of criminal proceedings selection.

**Figure 2 healthcare-11-01331-f002:**
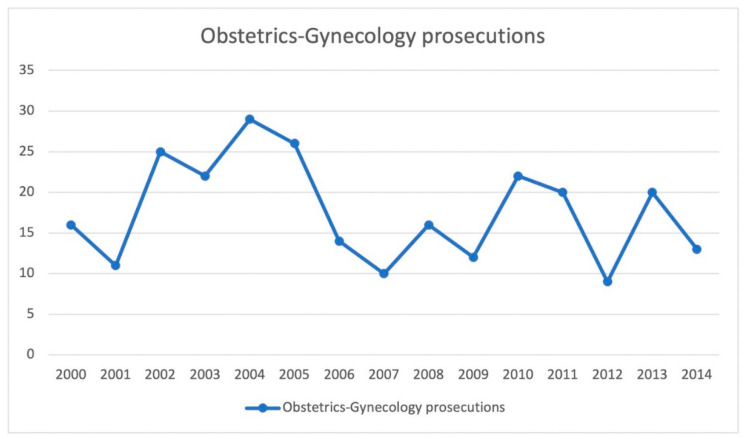
Trends in the number of medical malpractice cases involving the field of Ob-Gyn in Rome from 2000 to 2014.

**Figure 3 healthcare-11-01331-f003:**
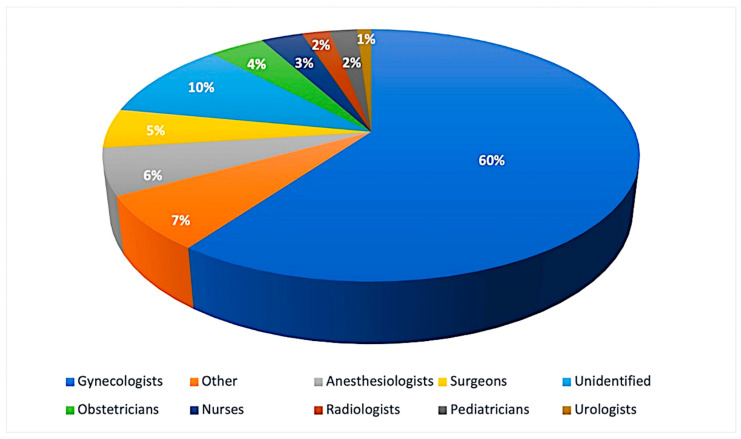
The number of specialists involved, subdivided by category.

**Figure 4 healthcare-11-01331-f004:**
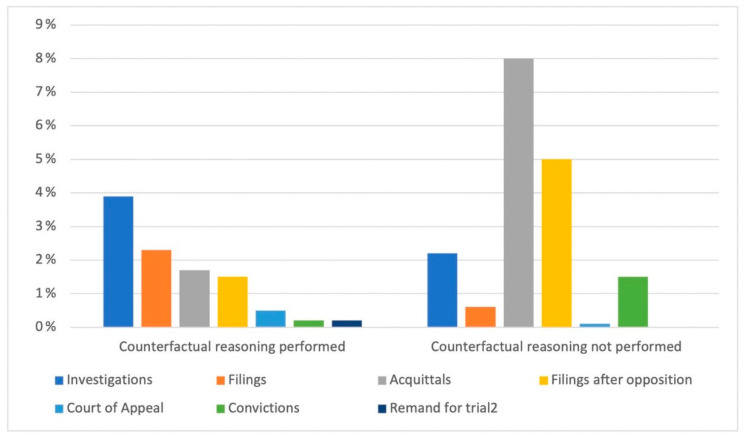
The distributions of the outcomes of criminal proceedings.

## Data Availability

Data are available upon reasonable request.

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
