# Peer review of "Medical Professional Liability in Obstetrics and Gynecology: A Pilot Study of Criminal Proceedings in the Public Prosecutor’s Office at the Court of Rome"

_healthcare, 2023, doi:10.3390/healthcare11091331_

Round 1
Reviewer 1 Report
Dear Author’s
I was pleased to review your article. The medico-legal aspects in present should change regulation in the majority of European countries.
Please change the figures using English.
It will be more educative if you add a flowchart with the studies.
Minor English Edits.
Author Response
Rev1Q1: Please change the figures using English. It will be more educative if you add a flowchart with the studies.
AA: We apologize for the oversight; all figures have now been translated into English language. We have also added a flow chart, as suggested by the Reviewer, to hopefully better clarify the steps followed in the selection of criminal proceedings.

Reviewer 2 Report
Dear Authors,
the work present a literally review and some considerations about liability in Gynaecology and Obstetrics.
Please, the article must be in english language. Not italian capture are needed. Before starting my review I ask you to modify tables and traslate all the text. please, when you talk about law , you should tell which order it belong to (italian penal code?), in order to make the reader understand you
happy to see your revised version in order to begin the review
Author Response
Rev2Q1: Please, the article must be in english language. Not italian capture are needed. Before starting my review I ask you to modify tables and traslate all the text. please, when you talk about law, you should tell which order it belong to (italian penal code?), in order to make the reader understand you.
AA: We thank the Reviewer and apologize for the oversight; all figures and text have now been translated into English language.

Reviewer 3 Report
Dear authors,
I have reviewed the manuscript entitled “Medical Professional Liability in Obstetrics and Gynecology: A Pilot Study of Criminal Proceedings in the Public Prosecutor's Office at the Court of Rome”.
My recommendations:
1-Although there were 13 cases in 2014, in Figure 1 it seems “0”. Please correct it.
2-Please do not use Italian in figures 2 and 3.
3-Provide high resolution images for figures.
4- In material and methods, The authors stated “The present study represents the evolution and development of a previous retrospective analysis conducted by the Institute of Forensic Medicine of the Catholic University of the Sacred Heart (Rome) which examined all proceedings related to medical professional liability within the Rome Public Prosecutor's Office's general register of crime reports, between January 1, 2000, and December 31, 2010.” Which study is this? Please use reference.
Author Response
Rev3Q1: Although there were 13 cases in 2014, in Figure 1 it seems “0”. Please correct it. Please do not use Italian in figures 2 and 3. Provide high resolution images for figures.
AA: We thank the Reviewer for the precise comments and apologize for the mistakes; we have now edited what was requested.
Rev3Q2: In material and methods, The authors stated “The present study represents the evolution and development of a previous retrospective analysis conducted by the Institute of Forensic Medicine of the Catholic University of the Sacred Heart (Rome) which examined all proceedings related to medical professional liability within the Rome Public Prosecutor's Office's general register of crime reports, between January 1, 2000, and December 31, 2010.” Which study is this? Please use reference.
AA: We apologize for not conveying the correct message and for the disorganized materials and methods section. We have now attempted to improve it and hope that it is adequate.

Reviewer 4 Report
Thank you very much for allowing me to review this article. The current work is about the evolution of cases of medical malpractice among obstetricians and gynecologists in Rome, Italy. The results of this study allow us to approach the impact that these cases have from a legal point of view.
I have several doubts and questions that I would like to be clarified in order to assess the publication of this work.
After reading the work several times, it is not clear to me whether it is a systematic review of the literature or a cross-sectional observational study in which the unit of analysis is the case being tried. If it is a systematic review, the articles included in the review and an interpretation of the findings of that review should be reported. If it is not a systematic review, I don't understand why it is reported in the methodology section. The methodology should be rewritten and it should be made clear what type of study has been carried out.
On the other hand:
Introduction: It should be rewritten to facilitate understanding. It should consist of only 3 paragraphs (background-current state of the topic-what is still unknown about the topic of study-study objectives). In its current format, there is no coherent structure.
Methodology:
The second paragraph of the methodology is part of the discussion. It is not methodology.
The type of study that has been carried out should be reported.
The study population and sample should be reported.
Reference should be made to the data collection tools and where the necessary data for the study was collected.
There is no statistical analysis section.
It should be made clear in this section that the legislation included in the study is Italian legislation (or so I believe).
It should be reported in this section that the study has met all the requirements of the ethics committee of their region/country.
Results:
The results that are in figures or tables should be separated and not repeated in the text. For example, the results that appear written in the first paragraph of this section are the same as those that appear in figure 1.
In figure 1, in the year 2014, 0 cases are reflected, and in the text, it is read that there were 13 cases. This information should be corrected.
The figures do not have good resolution. For example, figure 2 in the text (pdf) is very difficult to read.
The text boxes in the figures should be in English, not Italian. This should be corrected.
Discussion:
There are paragraphs in the discussion that are part of the results.
The first paragraph of the discussion should reflect the impact of the study's results on the main objective. It should be rewritten.
Conclusions:
The conclusions are not oriented towards the main objective or the title of the manuscript. The conclusions do not faithfully express the impact of the results. They can be understood as implications of the study, but they are not properly conclusions.
In the Institutional Review Board Statement section, the study registration code should be included, which is in the evaluator committee's file.
In the Informed Consent Statement section, I don't understand why reference is made to the donation of body or tissues before death if no samples or tissues from corpses have been used in the present study. From the methodology, it can be inferred that only data has been used. This should be clarified.
Thank you
Author Response
Rev4Q1: After reading the work several times, it is not clear to me whether it is a systematic review of the literature or a cross-sectional observational study in which the unit of analysis is the case being tried. If it is a systematic review, the articles included in the review and an interpretation of the findings of that review should be reported. If it is not a systematic review, I don't understand why it is reported in the methodology section. The methodology should be rewritten and it should be made clear what type of study has been carried out. The second paragraph of the methodology is part of the discussion. It is not methodology. The type of study that has been carried out should be reported. The study population and sample should be reported. Reference should be made to the data collection tools and where the necessary data for the study was collected. There is no statistical analysis section. It should be made clear in this section that the legislation included in the study is Italian legislation (or so I believe). It should be reported in this section that the study has met all the requirements of the ethics committee of their region/country.
AA: We agree with the Reviewer and thank him/her for the suggestion. The Materials and Methods section has now been edited and the sequence of the text in the manuscript has been adjusted with the hope that it will be clearer. In this section we also specified the means of data collection (paper files stored in the archives were accessed and analyzed, as well as electronic files using the Prosecutor's personal computer, whenever possible) and where the data were collected (General Register of Criminal Records of the Rome Prosecutor's Office). About the study's population and sample, we have included a flow diagram in the hope that it helps clarify the process of criminal proceedings selection. We did not devote a separate section to the statistical analysis because it was carried out solely using Microsoft's Excel platform. We have now specified all of the desired missing information.
Rev4Q2: Introduction: It should be rewritten to facilitate understanding. It should consist of only 3 paragraphs (background-current state of the topic-what is still unknown about the topic of study-study objectives). In its current format, there is no coherent structure.
AA: We understand the Reviewer's concern regarding the Introduction section and have attempted to reduce and improve the text.
Rev4Q3: The results that are in figures or tables should be separated and not repeated in the text. For example, the results that appear written in the first paragraph of this section are the same as those that appear in figure 1. In figure 1, in the year 2014, 0 cases are reflected, and in the text, it is read that there were 13 cases. This information should be corrected. The figures do not have good resolution. For example, figure 2 in the text (pdf) is very difficult to read. The text boxes in the figures should be in English, not Italian. This should be corrected.
AA: We believed it was important to discuss the statistical analysis results (which we portrayed in the figures) since the figures did not appear to be complete. We can remove the figures from the text if the Reviewer deems it necessary. We thank the Reviewer for the further comments; we have revised the figures.
Rev4Q3: There are paragraphs in the discussion that are part of the results. The first paragraph of the discussion should reflect the impact of the study's results on the main objective. It should be rewritten.
AA: We revised the discussion based on the Reviewer's suggestions; certain results from the analysis are presented in this section to allow comparison with the literature.
Rev4Q4: The conclusions are not oriented towards the main objective or the title of the manuscript. The conclusions do not faithfully express the impact of the results. They can be understood as implications of the study, but they are not properly conclusions.
AA: We changed the text in the hope that the adjustments made are adequate.
Rev4Q5: In the Informed Consent Statement section, I don't understand why reference is made to the donation of body or tissues before death if no samples or tissues from corpses have been used in the present study. From the methodology, it can be inferred that only data has been used. This should be clarified.
AA: We apologize for the mistake; we did not mean to add such information as it is not related to the topic discussed in the present research.

Round 2
Reviewer 2 Report
This is a retrospective study of criminal cases in obstetrics and gynecology in 2000-2014.
There is no scientific soundness. The debate on this topic is now superseded by new international juridical forms.
English do not follow journal's guideline. The manuscript needs extensive revision for language and grammar.
Author Response
Rev2Q1: This is a retrospective study of criminal cases in obstetrics and gynecology in 2000-2014.
There is no scientific soundness. The debate on this topic is now superseded by new international juridical forms. English do not follow journal's guideline. The manuscript needs extensive revision for language and grammar.
AA: We thank the Reviewer for his/her comment. We believe that our study may be of interest because, to the best of our knowledge, it is one of the few – if not the only – study in the literature that deals with professional liability prosecutions in Italy, specifically with reference to the Rome Public Prosecutor's Office, which is widely regarded as one of the largest in our country. As for the English, we agree with the Reviewer and would like to emphasize that the manuscript has undergone language revision through the MDPI platform.

Reviewer 3 Report
Thank you for revisions according to my recommendations.
Author Response
Thank you very much for you helping to improve the manuscript
Reviewer 4 Report
Thank you very much for allowing me to review your work. In my opinion, in this new version, the text has improved substantially. My contributions to the first version and the doubts I raised have been satisfactorily answered. I believe that in this new version, the text meets the standards to be published in this journal.
The only thing that I consider should be improved is the resolution of the figures that appear in his study. With the current resolution, it is difficult to read their texts, at least in pdf version.
Congratulations
Author Response
AA: We thank the Reviewer for the kind words in reference to our manuscript. As suggested, the figures have been changed and the resolution improved.
